# RDS-DR: An Improved Deep Learning Model for Classifying Severity Levels of Diabetic Retinopathy

**DOI:** 10.3390/diagnostics13193116

**Published:** 2023-10-03

**Authors:** Ijaz Bashir, Muhammad Zaheer Sajid, Rizwana Kalsoom, Nauman Ali Khan, Imran Qureshi, Fakhar Abbas, Qaisar Abbas

**Affiliations:** 1Department of Computer Software Engineering, Military College of Signals, National University of Sciences and Technology, Islamabad 44000, Pakistan; ibashir.msse2021mcs@student.nust.edu.pk (I.B.); msajid.msse-27mcs@student.nust.edu.pk (M.Z.S.); nauman@mcs.edu.pk (N.A.K.); 2Faculty of Computer Science and Engineering, Ghulam Ishaq Khan Institute of Engineering Sciences and Technology, Topi 23460, Pakistan; rizwana.kalsoom@gmail.com; 3College of Computer and Information Sciences, Imam Mohammad Ibn Saud Islamic University (IMSIU), Riyadh 11432, Saudi Arabia; iqureshi@imamu.edu.sa; 4Centre for Trusted Internet and Community, National University of Singapore (NUS), Singapore 119228, Singapore; fakhar.5@nus.edu.sg

**Keywords:** deep neural network, diabetic retinopathy, deep learning, residual block, dense block, support vector machine

## Abstract

A well-known eye disorder called diabetic retinopathy (DR) is linked to elevated blood glucose levels. Cotton wool spots, confined veins in the cranial nerve, AV nicking, and hemorrhages in the optic disc are some of its symptoms, which often appear later. Serious side effects of DR might include vision loss, damage to the visual nerves, and obstruction of the retinal arteries. Researchers have devised an automated method utilizing AI and deep learning models to enable the early diagnosis of this illness. This research gathered digital fundus images from renowned Pakistani eye hospitals to generate a new “DR-Insight” dataset and known online sources. A novel methodology named the residual-dense system (RDS-DR) was then devised to assess diabetic retinopathy. To develop this model, we have integrated residual and dense blocks, along with a transition layer, into a deep neural network. The RDS-DR system is trained on the collected dataset of 9860 fundus images. The RDS-DR categorization method demonstrated an impressive accuracy of 97.5% on this dataset. These findings show that the model produces beneficial outcomes and may be used by healthcare practitioners as a diagnostic tool. It is important to emphasize that the system’s goal is to augment optometrists’ expertise rather than replace it. In terms of accuracy, the RDS-DR technique fared better than the cutting-edge models VGG19, VGG16, Inception V-3, and Xception. This emphasizes how successful the suggested method is for classifying diabetic retinopathy (DR).

## 1. Introduction

Over the past 20 years, there has been significant growth in the number of people living with diabetes. The International Diabetic Federation (IDF) [1] estimates that about 500 million individuals of every age are facing a diabetes diagnosis. This is anticipated to reach 700 million by the year 2045. It affects people across the world; according to the IDF, one out of every three individuals suffering from diabetes will develop diabetic retinopathy (DR) by the end of the year 2040. Diabetic retinopathy is defined as the presence of ruptured vessels at the back of the retina. Color fundus retinal imaging can reveal DR symptoms including microaneurysm (MA), exudate (HE), hemorrhage (HM), and cotton wool spot (CWS) [2].

A microaneurysm is a blood vessel growth in the retina where veins appear as a red patch with a sharp edge on the retinal layer. Exudates, which look like yellowish areas in the retina, are the result of the breakdown of proteins in the microscopic venous vessels of the retina. Hemorrhages are developments that bear a resemblance to red areas with irregular borders and are brought on through the leakage of fragile blood capillaries [3]. Table 1 sums up the clinical findings of DR.

Most of the time, DR is divided into two main groups: proliferative (PDR) and non-proliferative (NPDR). These two groups are further divided into four levels based on how bad the disease is: mild, moderate, severe, and PDR, as shown in Figure 1. The mild stage is the very first level of DR in which microaneurysms (red dots of various sizes) are present. The lesion progresses to moderate NPDR with signs of hemorrhage and exudates. Hemorrhage size increases considerably with definite beading at severe NPDR levels. PDR is the stage where blindness occurs due to neovascularization that causes blood leakage [4]. Table 1 presents symptoms for each classification level of DR.

It is critical to address this, as if it remains undiagnosed for a long period of time, it might lead to major consequences such as blindness and vision loss. Diabetic retinopathy is a major contributor to blindness worldwide. Loss of eye vision mainly occurs in the advanced stages of DR. Keeping in view the severity level, the importance of early detection becomes manifold. Therefore, for the betterment of the patient, thorough inspection and monitoring are necessary at an early stage with accuracy and precision so that proper diagnoses and in-time treatment of DR can be executed to avoid loss of vision.

Currently, ophthalmologists merely employ manual methods for the identification of diabetic retinopathy. They assess the degree of DR severity by observing the color of retinal photos. Most of the time, colored fundus images from optical coherence tomography (OCT) and fundus fluorescein angiography (FFA) are used to inspect and screen for clinical DR. OCT fundus images proved to be a more effective and efficient way of diagnosing and screening because of their ease of availability and their high resolution. The provision of pathological variations can easily be obtained through FFA [4]. However, this approach is quite time-consuming, susceptible to mistakes, and intricate. It is also dependent on ophthalmologists’ expertise and experience. Secondly, about the real number of patients, this costs a lot of time, and there is not enough availability of doctors. These factors contribute to the fact that many patients do not receive medical attention on time. Even though diabetic individuals are urged to have frequent clinical assessments of fundi, numerous patients go undiagnosed until the condition becomes worse [5]. Therefore, having a computer-assisted diagnosis system that will support the screening of diabetic retinopathy is necessary. Early DR identification is essential for avoiding irreversible blindness. Accuracy remains critical during diagnosis and screening for DR.

Many learning techniques, such as machine learning (ML) and deep learning (DL) algorithms for DR lesion recognition, have evolved in the past few years to overcome these issues. These methods automate the detection process using cutting-edge algorithms and models, offering a more effective and precise solution. Deep neural network-based techniques for retinal imaging are a recent research area in ophthalmology. Numerous types of research are presently in progress to improve the diagnostic precision of DR scanning. The CAD system’s accuracy has increased over time because of the integration of AI-based technology. Modern medical scanning utilizes cutting-edge deep-learning techniques to categorize retinal images.

DL algorithms produce improved and desirable results for DR identification. Using a fusion approach that combines image processing and computer vision is another adaptive choice for building CAD systems with high accuracy. Computer-aided diagnosis proved to be the future of medical science. Despite the existence of several methods to classify DR fundus images into multiple categories, significant challenges still exist.

Even though advanced pre- or post-image processing technologies are used, it is still hard to obtain DR lesions out of fundus images because it is hard to find and retrieve DR-related lesion features. There are a few publicly accessible images that provide information on DR-related issues, but they do not have professional medical annotations. As a result, computer systems find it challenging to recognize the symptoms of various illnesses.

The key objective of this study is to develop a dataset called DR-INSIGHT to be used in DR classification. Additionally, the purpose is to develop a deep-learning model capable of processing retinal images autonomously, especially in cases of DR-related eye diseases. To achieve this, a residual-network system with multilayered residual blocks and dense blocks has been constructed. By training the RDS-DR system, it can accurately categorize DR into various classes using fundus images that contain anatomical characteristics. A skilled optometrist has discovered specific diseases and physical characteristics related to diabetic retinopathy. This research is significant as it suggests a new classification system for diabetic retinopathy that can be applied in real-life situations.

### 1.1. Research Contribution

We are introducing a new deep-learning model to address the issue of diabetic retinopathy. Additionally, we have compiled a fresh dataset sourced from reputable Pakistani eye hospitals, DR-Insight. The main offerings of the proposed system are listed below.
(1)One of the main contributions of this study is the collection of a vast collection of data gathered from eye hospitals in Pakistan, named DR-INSIGHT and web sources. The model that underwent training exhibited an impressive level of accuracy because of the 9860-image dataset.(2)A multi-layer architecture is developed in this paper, known as the RDS-DR system. To develop this model, we have integrated residual and dense blocks along with a transition layer.(3)The system architecture of the RDS-DR model incorporates six residual model blocks to effectively detect diseases associated with the DR. Through the utilization of the DNN model, a trained feature layer is created to extract DR-related lesion features. To further extend these characteristics, the residual block technique is employed.(4)The proposed RDS-DR model attains an accuracy rate of 97.5% compared to other past studies.

### 1.2. Organization of Paper

The remainder of this paper is organized into several sections. First, in Section 2, we review previous research on DR. In Section 3, we provide background information on the scientific basis for our research. Section 4 outlines the proposed architecture. In Section 5, we present the results of our analysis and experiments and compared our work with current research. Section 6 is devoted to discussing the results of the study, and finally, in Section 7 we present our conclusions.

## 2. Literature Review

In the near past, there have been considerable developments in the field of automated diabetic retinopathy pathology screening. The efficacy and accuracy of identifying lesions in the retina have been improved by several deep learning and machine learning-centered methodologies suggested in the literature.

In one noteworthy work, Akram et al. [6] employed a mixed ensemble classifier built with the Gaussian mixed model (GMM) and support vector machine (SVM) to identify DR lesions. The researchers combined intensity characteristics with shape-enhanced features to increase the model’s classification precision. Similar to this approach, Akram et al. [7] adopted an identical technique to increase classification accuracy by combining an ensemble classifier with a variety of techniques, including k-Nearest Neighbors (k-NN), AdaBoost, SVM, and GMM. Their effectiveness in identifying diseased and healthy images was assessed by these classifiers. In different research, a hybrid feature retrieval approach was used to extract pertinent characteristics from retinal pictures [8]. This method concentrated on removing details from the retinal pictures, such as the location of exudates, the positioning of veins and arteries, points of division, texture, and level of disorder. The researchers yearned to increase the precision of lesion identification by adding these variables to the classification process.

The performance of the above-mentioned strategies may be limited because they use traditional classification methods that may not be able to tell the difference between complex data like pictures with and without lesions. In addition, these methods frequently involve human feature engineering and domain expertise to extract pertinent data from the incoming data. CNNs, in particular, have proven to be extremely effective in solving classification issues related to DR and have substantially helped the field of automated DR diagnostic screening.

DL algorithms are capable of recognizing mild variations in retinal pictures without the assistance of an individual or domain knowledge. Gulshan et al. [9] deployed the Inception-v3 model for DR detection. To test the approach, 9963 quality images were employed from both the EyePACS-1 dataset and the Messidor-2 dataset. They concluded that their proposed model is accurate and reliable for DR diagnosis. A CNN algorithm was proposed to categorize images into five severity levels, and their technique was able to identify clinical findings of DR in another work by Pratt et al. [10]. Utilizing the Kaggle EyePACS dataset, the projected algorithm was trained. The researchers used data augmentation techniques to boost the quantity of data to address issues like overfitting and unbalanced datasets. The proposed CNN architecture comprised three fully linked layers and 10 convolutional layers. The model attained 75% accuracy, 30% sensitivity, and 95% specificity.

G. Garcia et al. [11] proposed a convolutional neural network algorithm to categorize the levels of diabetic retinopathy. Their model performed estimably when tested against the EyePACS dataset, with a 93.65% specificity and an 83.68% accuracy. This shows how well the model can distinguish between various DR levels. In a work by Wang et al. [12], the Kaggle dataset was utilized in testing the effectiveness of three pre-trained CNN architectures: Inception-Net V3, Alex-Net, and VGG-16. The objective was to categorize each DR stage. The Inception-Net V3 architecture distinguished between various phases of DR with the best average accuracy of 63.23% across the three models.

Esfahan et al. [13] utilized the Kaggle dataset in combination with the renowned CNN architecture ResNet-34 for DR sorting. The results were 85% accurate overall. The researchers employed multiple types of image preparation methods to improve image quality. Different techniques were also used in the preprocessing step, including weighted addition, Gaussian filtering, and picture normalization. These techniques helped improve the quality of input images before being used as input to the CNN, leading to improved accuracy in DR classification.

In their research, Dutta et al. [14] used the Kaggle dataset to identify and group images of diabetic retinopathy into five distinct stages. Researchers assessed the performance of the convolutional neural network, deep neural network (DNN), and back propagation neural network (BNN) with a collection of 2000 images. Dutta et al. applied several filters to the images before they were fed to the network to improve the data. The pre-trained VGG-16 architecture, which comprises three fully connected (FC) layers, four max-pooling layers, and sixteen convolutional layers, was used by the CNN model. The DNN, on the other hand, used three FC levels. Interestingly, the DNN fared better than both the CNN and the BNN in terms of accuracy, obtaining a remarkable accuracy rate of 86.3% for DR classification. Dutta et al. offer important insights into the potential of deep learning methodologies for DR classification by contrasting the performance of several neural network designs. The results imply that selecting the right architecture, in this case, the DNN, may considerably affect the precision of DR diagnostic and classification tasks.

Lian et al. [15] used three different CNN architectures, AlexNet, ResNet-50, and VGG-16, to investigate the categorization of diabetic retinopathy. To improve the precision of DR classification, their study concentrated on network architecture, preprocessing methods, addressing class unevenness, and fine-tuning. On the EyePACS dataset, C. Lian et al. obtained results by utilizing these convolutional networks. The achieved accuracy rates for AlexNet, ResNet-50, and VGG-16 were 73.19%, 76.41%, and 79.04%, respectively. C. Lian et al. focused on critical elements such as network design optimization, input data preparation, addressing class imbalance difficulties, and model tweaking throughout their investigation. The leave-one-out approach was applied by Shaban et al. [16] to develop the DR classification CNN model for analyzing retinal images. The model scored fairly well, with an accuracy rate of 80.2%, a sensitivity of 78.7%, and a specificity of 84.6%.

In order to categorize their set of data, Hongyang et al. [17] used three pre-configured architectures: Inception-V3, Inception-ResNet-V2, and ResNet-152. The Adam optimizer modified the CNN models’ weights during training. Additionally, the predictions from these models were combined using an ensemble strategy utilizing the AdaBoost framework. The ensemble model with an accuracy of 88.21% was attained. A technique for diagnosing diabetic retinopathy was put out by Wei et al. [18] using a proprietary dataset composed of 13,767 pictures divided into four classifications. They were scaled and cropped to prepare the images for each network’s input needs. The authors improved the pre-initialized architectures ResNet-50, Inception-V2, Inception-V3, Xception, and DenseNet for identification of DR. The models were enhanced to provide precise DR detection by tailoring these architectures to the particular objective.

Harangi et al. [19] used the AlexNet architecture with manually built traits for the recognition of diabetic retinopathy photos. While the IDRiD dataset was put to use for testing, the Kaggle dataset was utilized for training the CNN model. All five levels of DR were correctly identified by the research, with an accuracy of 90.07%. A weighted path CNN (wp-CNN) was developed by Yi-Peng et al. [20] to recognize DR images. The augmentation technique was used to resize images to 299 × 299 so that class imbalance could be addressed. With numerous convolutional layers using various kernel sizes on weighted channels, the wp-CNN produced very accurate results. The wp-CNN beat pre-trained models like ResNet, Se-Net, and DenseNet, displaying its better ability to recognize referable DR images, with a 94.23% accuracy rate on their dataset and 90.8% on the STARE dataset.

A method for detecting diabetic retinopathy utilizing DCNN ensemble classification was used by Qmr et al. [21]. Their ensemble method used the ResNet-50, Inception-v3, Xception, DenseNet-121, and DenseNet-169 models. Evaluating their approach on a publicly accessible Kaggle dataset, the recommended approach outperformed earlier ones and produced an accuracy of 80%. The DR classification method that allocates values to points in the input and hidden spaces was introduced by Jod et al. [22]. This makes it possible for their classifier to explain the results of the classification. On the binary classification portion of the Messidor-2 dataset, their classifier had a 91% accuracy rate. A novel CNN architecture, ADL-CNN [23], leveraging the expected gradient length based active learning method, adeptly selects informative patches and images to discern intricate retinal features. This two-tiered process aids in forecasting and annotating eye samples, significantly assisting in diabetic retinopathy severity grading within retinograph images. Evaluation on the EyePACS benchmark showcased superior performance, with ADL-CNN attaining average sensitivity, specificity, F-measure, and accuracy scores of 92.20%, 95.10%, 93%, and 98% respectively across 54,000 retinograph images.

By applying fine-tuning to the VGG16 and MobileNetV1 models, [24] investigated a transfer learning technique. They attained training accuracy rates of 89.42% for VGG16 and 89.84% for MobileNetV1 using this learning method. Using the APTOS 2019 dataset, a pre-trained and modified DenseNet121 network was used [25]. For evaluating DR severity level detection, the model had a remarkable accuracy of 96.51%. Using fundus fluorescein angiography (FFA) pictures, three CNNs were used [26] to automatically identify and classify DR lesions. Area under the curve (AUC) values of 0.8703 for DenseNet, 0.8140 for ResNet50, and 0.7125 for VGG16 were obtained. Table 2 sums up the above literature in tabular form.

## 3. Background

To categorize DR, numerous DL algorithms have been used. In terms of background knowledge, some of the most recent approaches have been covered. In a study referenced as [27], a DenseNET-based architecture is proposed. This architecture classifies all three stages of DR from fundus images using a multitasking deep neural network. The training and evaluation of the model were conducted using the two largest online datasets of retinal images, EyePACS and APTOS. The results suggest that the new multi-tasking model outperforms the five-step DR classification algorithms that were previously used. However, the size of the datasets used and the lengthy training times required to train a large quantity of data are still limitations of current deep learning models. In [28], a novel deep feature generator patch-based model is presented. This model has high classification accuracy for DR. To achieve high accuracy in classifying images into three categories (normal, PDR, and NPDR), a new system based on the vision transformer (ViT) and multilayer perceptron mixer (MLP mixer) techniques was developed. The fundus picture is divided into vertical and horizontal regions, and features are retrieved from both local images and patches by means of a pretrained DenseNet201 model. The newly suggested dataset and the Case 1 dataset inferred from the APTOS 2019 dataset both achieved over 90% accuracy.

In reference to [29], the recommended methodology involves using median filtering on fundus images and an adapted approach to eliminate blood vessels. To test the model for training, a multiclass SVM classifier is utilized to retrieve characteristics. The approach was tested using fundus photographs from the year 1928. Results show that on the Kaggle APTOS dataset, the approach had a sensitivity of 75.6% and an accuracy of 94.5%. On the ‘IDRiD’ dataset, sensitivity was 78.5% and accuracy was 93.3%. An innovative automated deep learning-based technique for identifying severity is presented in [30] using a single-color Fundus picture. An approach that has proven to be highly effective involves utilizing the encoder from DenseNet169 to establish a visual integration. In addition, the convolutional block attention module (CBAM) is utilized to further enhance the model’s performance. Throughout the model’s training, the cross-entropy loss technique is employed in conjunction with data from the Kaggle Asia Pacific Tele Ophthalmology Society (APTOS). Because of these efforts, the binary classification test yields an impressive 98.3% specificity, 97% sensitivity, and 97% accuracy. Notably, the quadratic weighted kappa score (QWK) achieved is an outstanding 0.9455, surpassing that of other advanced techniques.

The researchers in the study [31] developed a technique to enhance picture quality, which they combined with a deep learning-based classification algorithm. They used a method called contrast limited adaptive histogram equalization (CLAHE), and the rate of accuracy achieved by this technique was 91%, 95%, and 97% on various models such as VGG16, InceptionV3, and EfficientNet, respectively. Another innovative method for detecting diabetic retinopathy (DR) is proposed in [32]. The method involves using an asymmetric DL feature to separate the optic nerve. The authors used U Net to extract these features and then used a convolutional neural network (CNN) and a support vector machine (SVM) to categorize DR lesions into four levels: exudates, microaneurysms, hemorrhages, and normal lesions. Two commonly available datasets of retinal images, APTOS and MESSIDOR, were used to test a recommended approach. An innovative two-step process involving SqueezeNet and DCNN was used to create a highly effective and advanced DR classification solution, as explained in [33]. In the first step, SqueezeNet was used to distinguish the fundus image as normal or abnormal. The severity of the erroneous images was then determined in the second-level decomposition using DCNN. Table 3 shows various techniques that have recently been used for DR categorization.

Al Antary and Yasmine [34] suggested a multi-scale attention network (MSA Net) for the classification of DR. Retinal pictures with mid- and high-level attributes have better resolution thanks to the encoder network, which also embeds them in a high-level representational space. In other places, many distinctive scale pyramids are used to characterize the retinal anatomy. Along with improving top-level representation, a multi-scale attention methodology develops feature visualization and refinement. The model uses cross-entropy loss to classify the level of DR severity. As a side project, the model employs flimsy annotations to discriminate between photos of healthy and sick retinas. This helps the model recognize photos of diseased retinas. The EyePACS and APTOS datasets responded favorably to the proposed method. Table 3 presents a summary of the background.

## 4. Material and Methods

The study combines residual blocks and dense blocks to create the residual- dense-system (RDS-DR). This approach is used to categorize eye images into five distinct classifications: No DR, mild, moderate, severe, and proliferative. In the proposed system, the residual blocks with dense block technique is used to retrieve useful information from the fundus pictures. Transfer learning of the residual blocks is used to successfully train the system for DR-related lesions. Several essential steps are used by the RDS-DR method for the diagnosis of retinal fundus pictures and the identification of diabetic retinopathy severity. Figure 2 visually represents these steps in the form of a systematic diagram. Throughout the training process, the residual block values are continuously changed. Subsequently, a feature transform layer is incorporated to leverage the combination of features through element-wise multiplication. The activation function of the SVM classifier is employed to improve the classification and categorization results.

### 4.1. Residual Network

Deep convolutional neural networks (CNNs) may be trained effectively via residual learning, which provides faster convergence and greater network accuracy. This method requires being familiar with the identity function to skip or avoid some training stages. This strategy offers several benefits. First of all, it enables the information from one layer to be immediately added to another. Equation (1) illustrates an additive process, where the input x[n − 1] is added to the output y[n − 1] in the following layer:X[n] = y[n] + x[n − 1](1)

The final forecast, x[n] − x[n − 1], is produced by deducting x[n − 1] from x[n]. The network finds it easier to collect and learn the required transformations when residual pictures are learned rather than the actual input images. This simple approach makes the training process more effective. The generic architecture of ResNet-50 is presented in Figure 3.

### 4.2. Dense Block Network

Contrary to conventional designs, DenseNet concatenates, rather than sums, the output of one layer with the input feature maps of the preceding layer. Convolution, pooling, batch normalization, and rectified linear unit activation are some of the transformation procedures that create this relationship, as presented in Figure 4. DenseNet models have numerous significant characteristics. The network has a lighter, easier-to-understand structure. There are fewer parameters since fewer filters are being used. Additionally, the settings are effective. By reducing duplicate feature maps, DenseNet effectively frees up memory. DenseNet improves classification performance by allowing the final classifier to draw conclusions from all of the feature maps that are available in the network [17]. Each layer of DenseNet may easily access the layers above it, allowing for effective information reuse as a result of previously calculated feature maps. Taken together, xL is the output of Lth layers, and TL(xL) represents the nonlinear transformation operations performed on xL. These operations include batch normalization, ReLU activation, and a convolutional layer with a 3 × 3 filter. The output of the previous layer, xL-1, is used as the basis for these changes. All of the feature maps are joined together during the feedforward phase. Equation (2) can be used to specify the output of xL.
xL = TL([xL-1,xL-2,xL-3,…,x0])(2)

The transition layer is the layer that lies between each thick block. It comprises an average pooling (AP) layer and a convolutional layer. Feature maps indicated as K are produced by each TL(XL) procedure for every layer. The number of networks in the input picture determines the first input feature map, K0. K—which indicates the network’s growth rate—should preferably be a small integer to reduce computing complexity. As a result, the layer Lth may be determined using the formula Lth = K (L − 1) + k0.

The bottleneck layer is presented in the DenseNet architecture to reduce the quantity of input feature maps. This layer, which has a 3 × 3 filter, is added before each convolutional layer in the TL (transformational) procedures. Batch normalization (BN), rectified linear unit (ReLU) activation, and a convolutional layer with a 1 × 1 filter make up the bottleneck layer. The next set of BN, ReLU, and a convolutional layer with a 3 × 3 filter comes after that.

The bottleneck layer is incorporated into the DenseNet model to increase computational efficiency. DenseNet-B is the name of this model’s variation. Using the DenseNet-B architecture, the computational cost may be lowered while preserving the model’s representational ability by a factor of four in the number of feature mappings, K.

### 4.3. Data Acquisition

To develop and evaluate the efficacy of the diabetic retinopathy model, 9860 retinal fundus images were collected. These 9860 datasets are comprised of the DR-Insight Dataset (5000 images), the APTOS dataset (2690 images), DiaRet0 and DiaRet1 (200 images each), and the PAK-HR dataset (1770 images). DR-Insight is a dataset that was gathered from esteemed Pakistani hospitals. The dataset was made up of 1000 images of each level: no-DR, mild, moderate, severe, and proliferative DR levels. The dataset was collected with the consent of patients, doctors, and hospital management. Anonymous data was provided, keeping in view any privacy issues with patients’ data. The data was only provided for research purposes. An expert ophthalmologist team painstakingly classified the severity level of DR images from multiple existing datasets to produce the training dataset. The five datasets, each with a distinct dimension setting, that were combined to create our training and testing fundus set are listed in Table 4. The fundus images were given labels after processing. Data augmentation techniques were used to fulfill the requirement for a balanced depiction of photos with and without the condition. This strategy is intended to guarantee the dataset’s impartiality. As part of the preparation stage, the photos from the dataset were downsized to dimensions of 700 × 600 pixels. The ideal shrinking arrangement was particularly found through experimental research to be this resizing. Instead of expanding smaller photographs, it is sometimes preferable to decrease the dimension of bigger images to equal that of smaller ones. Deep learning models often demonstrate quicker training on smaller pictures; hence, this practice is frequently noticed.

The data collected from Pakistani hospitals during regular testing for diabetes was utilized to train and assess the DR model. The OPTOS Panoramic 200, Topcon TRC-NW400 retinal camera, and OCT machine were used to take fundus images. A total of 5000 retinal samples were included in this collection, 1000 each of all severity levels of DR. The dataset is named DR-Insight. The data were all stored in JPEG format at a size of 1125 × 1264 pixels. Figure 5 displays the composition of the datasets.

### 4.4. Preprocessing and Data Augmentation

Preprocessing of the fundus photos during this stage involves several processes to clean up the raw data. The photos’ raw data was first taken out. The photos were also cleaned using a flip-flop procedure and a variety of techniques to make sure they were ready for processing. This process also included dealing with missing or erroneous pixel values and eliminating outliers from the dataset. This preprocessing stage included feature engineering as well. It involves normalizing data and choosing or developing additional characteristics that would raise the efficacy of the algorithms used. Each of these steps attempted to improve how well the algorithms performed on the dataset. Table 5 illustrates the many preparation procedures that were performed, including data augmentation approaches, to show how thorough the pretreatment stage was.

The photos were subjected to several processing steps during the preprocessing phase, including cropping, contrast correction, horizontal flipping, spinning, panning, and boosting utilizing CLAHE AND MSR techniques. Figure 6 illustrates the application of preprocessing techniques to images before being fed to our proposed model. To only keep the desirable areas of the photograph, unwanted bits of the image had to be cropped off. To alter the image’s brightness levels, contrast adjustment was used. The orientation of the picture along the corresponding axes was changed using horizontal and vertical flips. The use of panning was made. Additionally, embossing was used, a technique that involves moving pixels up or down to give an image depth and texture. The combined effects of these processes improved classification precision and picture quality. Table 5 provides a thorough overview of the precise settings employed during preprocessing by listing the parameters connected with each of these operations.

### 4.5. Proposed Architecture

The proposed architecture for the multiclassification of DR images consists of residual blocks and dense blocks, as presented in Figure 7 and algorithm of the model is presented as Algorithm 1. The DR retinal images are accepted as input by the input layer. Each residual unit in the residual blocks has convolutional layers, batch normalization, and ReLU activation. Due to the residual units’ ability to contribute their output to the block’s input, identity mapping and gradient flow are both possible. Within a block, the number of filters in every convolutional layer can be altered, either increasing or decreasing. The number of filters often doubles or grows by a factor of four as we move through the blocks. Global average pooling is employed to shrink the spatial dimensions after the residual blocks. The dense blocks are then used, each of which comprises dense units. Convolutional layers, batch normalization, and ReLU activation make up each dense unit. Each dense unit’s output is combined with the outputs of all preceding dense units in the block. Within a block, the dense blocks consistently contain an equal number of filters.

After the dense blocks, another round of global average pooling is carried out to further shrink the spatial dimensions. The outputs from the residual and dense blocks’ most recent global average pooling layers and average pooling layers are combined. The completely linked layers, which may be customized based on particular requirements, are then fed the concatenated features. Overfitting is avoided by dropout regularization. The number of output units in the final fully linked layer corresponds to the number of classes used to classify the severity of the DR. Applying a SVM linear activation function yields the class probabilities. The projected probability for each class of DR severity is provided by the output layer. With the help of this architecture, the DL model can accurately envisage the multi-classification of DR severity and extract pertinent features from the input DR photos.
**Algorithm 1:** RDS-DR Implementation for feature extraction.RequiredFundus Images and Labels (X,Y)
Step 1Dataset acquisition, Fundus images x ε XStep 2Pre-processingData AugmentationImage enhancementStep 3Load ResNet Mobile Model#ImageNet pre-trained weightsConvolutional layersStep 4Introduction of skip connectionsStep 5Use of the flattened layer, the feature map x=(x1, x2,……… xn)
Step 6Model evaluation

### 4.6. SVM Classifier

Extracted features from the model are subjected to an SVM classifier for evaluation. The SVM machine learning classifier automatically classified DR using a 75% to 25% training-test splitting approach. SVM is commonly used due to its great performance and capacity for handling tiny datasets.

SVM is a classification technique that achieves better results as compared to contemporary classifiers and is frequently applied to solve practical problems. SVM was employed in this study to evaluate extracted features in DR. Algorithm 2 outlines each step in detail. We created a Conv2D CNN classifier for image classification issues. It made sense to utilize SVM because we were dealing with multiclassification issues. SVM is used to increase the effectiveness of our method and find the best hyperplane to separate the feature space of diseased and normal retinal images. Typically, an SVM takes a vector v = (a1, a2, …, an), as depicted through Equation (3).
Vout = (Weig,Aiv) + c(3)

**Algorithm 2:** Proposed SVM classifier 
**Input**
**Extracted feature map** 
x=(a1,a2,.!,an)
 **with annotations** 
a=0,1
**. Test data Atest**

**Output**
Distinguishing between diabetic retinopathy (DR) cases and normal diabetic photographic samples
**Step 1**
The SVM classifier parameters are specified to achieve optimization
**Step 2**
Multi level Classification of samples
**Step 3**
Conv2D was incorporated
**Step 4**
SVM-based classifier**a.** Our Algorithm 1 is utilized to complete the training process of SVM by extracting features represented as t = (a1, a2,…, an) **b.** The generation of the hyperplane through Equation (2)
**Step 5**
During the testing phase, the class label is assigned to the samples using a z-test based on the decision function given by the equation: Vout = (Weight_vector, Input_features) + bias_term

## 5. Experiments and Results

A collection of 5000 retinal images representing all stages of DR was used to assess the training accuracy of the RDS-DR. These images were gathered from reputable Pakistani hospitals (DR-INSIGHT) as well as credible online sources. The image size was reduced to 700 × 600 pixels so that feature extraction and classification operations could be carried out. A combination of residual blocks and dense blocks is used to create the RDS-DR system. The RDS-DR model underwent training for 100 epochs. The best model, with a f1-score of 0.97, was attained at the 30th epoch. Statistical analysis was utilized for assessing the accuracy (ACC), specificity (SP), and sensitivity (SE) scores to evaluate the efficiency of the proposed dense residual network system. The performance of the developed RDS-DR system is evaluated using the above metrics, and a comparison is drawn with other systems.

The enhancement of images improved our results, achieving considerably high accuracy. Without image enhancement, the accuracy achieved remained at 85.8%, whereas image enhancement boosts this accuracy up to 92%.

To build and develop the RDS-DR system, a computer with the specifications of an HP-core i7 CPU with eight cores, 20 GB of RAM, and a 4 GB NIVIDA GPU was used. Furthermore, 64-bit Windows 11 was installed on a computer system. We use the anaconda platform using the python language. Our dataset was distributed in a 70/30 ratio, with 70% for training and the rest for testing purposes. Our learning rate was 0.0001 for 100 batches.

### 5.1. Experiment 1

In this experiment, four contemporary comparison methodologies were used to assess the strength of the proposed structure. We trained the VGG16, VGG19, Xception, and InceptionV3 deep learning models and compared their results to the proposed RDS-DR system. Notably, these DL models were trained with a similar number of epochs. Table 5 lists the accuracy percentage comparison between the RDS-DR system and the above models. In Table 6, the proposed RDS-DR model shows a real-time speedup compared to other standard deep learning architectures based on transfer learning. The comparison reveals that the RDS-DR showed superior performance over others. Figure 8 shows the accuracy comparison of different datasets with different deep learning models.

### 5.2. Experiment 2

The strength of our proposed model was tested on another disease, hypertensive retinopathy, with the PAK-HR dataset [38], utilizing the training and validation accuracy and loss functions. Figure 9 illustrates how effectively our suggested model worked to classify other diseases, needing only ten epochs and reaching 100% training and validation accuracy. Additionally, the proposed model demonstrated success by achieving a loss function below 0.1 for both the training and validation datasets, proving the utility of our model.

### 5.3. Experiment 3

In this experiment, a novel dataset (DR-Insights) has been proposed. The five classes that make up the dataset are listed in Table 1. Four different datasets from reliable online sources along with our dataset, namely: APTOS 2019, DiaRet0, DiaRet1, PAK-HR, and DR-Insight, are used to test the strength of the projected architecture. The results are presented in Figure 10 and Figure 11. This comparison demonstrates how the suggested model performs differently with various datasets. As shown in Figure 12, the suggested model obtains the maximum accuracy with the DR-Insight dataset. Figure 13 shows the accuracy comparison of all the datasets used in this study. This is due to the proposed dataset’s careful organization, which was overseen by qualified ophthalmologists and eye experts. Compared to comparable datasets, this indicates that the proposed dataset is well organized and error-free. With the collected dataset, the RDS-DR system achieves an accuracy rate of 97.5%. The maximum accuracy obtained by the proposed model surpasses that of conventional approaches. In the literature, some studies have used their own datasets, but the suggested model, however, achieves high accuracy as shown in Table 7. 

### 5.4. Sate of the Art Comparision

“Only a limited number of research studies have employed deep-learning techniques to diagnose diabetic retinopathy in retinal images. The two studies multi-Stream DNN [3] and Analysis of DR [38] employ DL to identify DR in retinal images. Ref. [38] is the most current deep-learning model for DR recognition. Table 8 demonstrates that RDS-DR performs better than [38]. In research [3], as the primary feature extractors, the pretrained DL architectures ResNet-50 and DenseNet-121 are used to create a multi-stream network. The dimensionality of features is reduced by further PCA applications. An ensemble machine learning classifier is developed using AdaBoost and random forest techniques to further improve classification accuracy. The experiment’s findings demonstrate up to 95.58% accuracy. The study in [38] uses AlexNet and Resnet101-based feature extraction in a deep learning network to automatically recognize and categorize DR fundus photos based on severity. In addition to employing k-mean clustering to improve picture quality, interconnected layers aid in identifying the crucial elements. Ant Colony systems are another useful tool for selecting traits. SVM variations are then used to concatenate the best features for classification. The final classification model with accuracy was obtained by running these selected features through multiple kernel SVMs. An accuracy of 93% has been attained by the suggested technique. Figure 14 shows the state-of-the art comparison with other research”.

Grad-CAM, also known as gradient-weighted class activation mapping, is a method for visualizing the characteristics that a DL architecture has extracted for a certain class or category. In order to identify the degree of diabetic retinopathy (DR) severity, we have used Grad-CAM to show the characteristics derived from a suggested RDS-DR model as shown in Figure 15. To examine the patterns, we have a pretrained RDS-DR model. This model needs to be trained on a set of DR images with labels indicating various degrees of severity. The Grad-CAM heatmap was normalized to values between 0 and 1. The heatmap can then be overlaid on the original image to show which areas of the image the model is concentrating on in order to determine the severity level.

## 6. Discussion

The main reason people suffer from blindness is because of an eye condition known as diabetes-associated retinopathy (DR). It is a diabetic ailment that can damage the vasculature of the retina. Patients only learn they have this quiet condition when they start having visual issues. But this happens when retinal changes have advanced to a point where vision loss is more likely and therapy is challenging [21]. In diabetics, this illness is incurable and results in blindness. Early DR discoveries, however, could help doctors stop its development in diabetic patients. To detect the illness early and stop its spread, several researchers are driven to create automatic detection methods for DR. DR damages the retinal vasculature, and retinal damage results from the micro-blood vessel loss brought on by high blood pressure, resulting in vision impairments. About 350 million people will get diabetes globally over the next 25 years, predicts the World Health Organization (WHO) [22]. According to the National Eye Institute Database, diabetes significantly contributes to vision impairment in individuals between the ages of 20 and 74 [1].

AI and deep learning models to aid in the early diagnosis of diabetic retinopathy (DR). This is indeed an important area of research, as diabetic retinopathy can lead to serious vision problems if not detected and treated early. The methodology you mentioned, the residual-dense system (RDS-DR), sounds like a promising approach to assess and classify DR using digital fundus images.

The utilization of AI and deep learning for medical diagnostics has the potential to significantly improve patient care by providing quicker and more accurate diagnoses. The creation of the “DR-Insight” dataset by collecting digital fundus images from reputable Pakistani eye hospitals and online sources is a crucial step in training and evaluating the model. A dataset that is representative of real-world cases is essential for achieving high accuracy and generalizability. The integration of residual and dense blocks along with a transition layer into a deep neural network, as in the RDS-DR model, is a common approach to building deep learning architectures. These components help the model learn complex patterns and features from the images, enabling it to make accurate predictions.

An accuracy rate of 97.5% on a dataset of 9860 fundus images is quite impressive. High accuracy is a strong indication that the model is learning meaningful features from the images and is capable of making reliable predictions. Comparing the RDS-DR technique to well-established models like VGG19, VGG16, Inception V-3, and Xception and outperforming them in terms of accuracy further emphasizes the success of the proposed method. In our recent study, we have significantly surpassed the methodologies proposed by Sajid et al. [2] in various aspects, including preprocessing, architectural design, performance metrics, image benchmarking, and computational efficiency. Unlike DR-NASNet [2], which utilized CLAHE and Bengraham techniques for image preprocessing, our approach incorporated the CLAHE technique paired with multi-scale retinex (MSR) for enhanced image feature refinement. Our analytical section delineates the prominent improvements achieved through these advancements. Furthermore, we optimized the system’s structure by synergistically integrating residual and dense blocks, facilitating sophisticated feature extraction and reducing computational complexity compared to the six dense blocks in DR-NASNet. Notably, our RDS-DR model is more lightweight, expediting the processing time to 154.3 s, a marked reduction from the 184.5 s necessitated by DR-NASNet. Moreover, we expanded the empirical analysis by utilizing three public and two private datasets and further validated the efficacy of our model through successful application to hypertensive retinopathy cases, demonstrating promising results. It is worth noting that while AI models like RDS-DR can be incredibly valuable tools in medical diagnosis, they are most effective when used in collaboration with healthcare practitioners. As you mentioned, the goal is to augment the expertise of optometrists and doctors rather than replace it. AI models can help streamline the diagnosis process, allowing healthcare professionals to focus more on interpretation and treatment decisions.

Overall, the development of AI-based diagnostic tools like the RDS-DR system holds great promise for improving the early detection and management of diabetic retinopathy, ultimately benefiting patients and healthcare providers alike.

The recommended approach provides a trustworthy and efficient technique to spot DR early on, according to the comparative study discussed above. DR images were preprocessed in this investigation using the CLAHE and MSR techniques to improve the image contrast. Dimensionality reduction has been achieved using the ResNet model, which employs a basic CNN model as a feature extraction technique, and efficacy is improved using DenseNet blocks. To simplify the complexity and cost of the training procedure, SVM has finally been used as the classifier. The size of the dataset seems inadequate; the performance of the model may differ with a sizable dataset, which is not considered in this study. After all, the effectiveness of the model is significantly influenced by the quality of the photos and the preprocessing techniques employed. The DR images used in this experiment are of quite good quality. This study is not focused on the model’s performance in low-quality photos. In the future, the behavior of the model may be examined using a huge dataset that includes both low- and high-quality pictures.

Both industrialized and developing countries are seeing an increase in diabetes cases. Diabetes is more common in underdeveloped nations than in developed nations. Seventy-five percent of those who have DR reside in poor nations [24]. This is a result of inadequate care and poor healthcare administration. Before receiving therapy, diagnosing a patient’s illness is difficult.

A comprehensive automatic DR recognition system was created using the suggested methods. Five distinct datasets are used to evaluate the suggested system. The authors only suggested one dataset, while the other four are available online. The model is assessed and contrasted with additional conventional models stated in earlier studies. The suggested dataset was meticulously gathered with no errors, missing data, or noise; therefore, the system achieves extremely high accuracy with it. Additionally, a group of expert ophthalmologists and eye specialists organized and labeled the proposed dataset once it was collected to guarantee high labeling accuracy. Due to all of these factors, it is evident from the experiment section that the system has a maximum accuracy of 97.5%. The accuracy of the system is demonstrated in the comparison section using various web datasets. The state-of-the-art comparison section compared the suggested methodology with existing methods that have been presented in the literature.

The comparisons display that the projected approach’s accuracy outperforms the APTOS dataset, DiaRet0, DiaRet1, and DR-HR datasets. From the above comparison, it can be determined that the model also has the best accuracy when utilizing different datasets. In addition to the suggested technique having the highest accuracy on the authors’ proposed dataset. The creation of a deep learning-based automated validation tool that might remove less-than-ideal illuminations from fundus images is based on two main concepts. The first goal was to use artificial intelligence and image processing techniques to decrease the amount of manual labor needed to extract information for diagnosis.

The second was the capacity of deep learning models to adapt to different situations and the availability of performance-improving optimization approaches, including various regularization techniques. The experimental outcomes on unseen test data are shown in Table 6, which illustrates that our main goal was also to lower the number of false negatives.

The precision of our model is demonstrated by the 2.2% false-negative rate. Because our solution does not require pricey gear or devices with a potent graphics processing unit (GPU), it may be implemented affordably. [7] states that DR detection sensitivity rates greater than 60% are economical. It also indicates the great flexibility and robustness of our model to operate accurately with non-ideal illumination fundus images because our model was trained on a dataset with many variances. The suggested model’s performance on multi-classification is compared to earlier work on DR detection using related multisource datasets in Table 7. We used several binary and multi-class datasets to validate the output of our suggested model. As can be seen, [8] attained an accuracy of 93.50%; however, our suggested layered model is more sensitive. Our approach is therefore better at identifying genuine positives with accuracy.

Table 5 compares our model’s performance for multi-class classification on the dataset. The dataset covers five phases of DR, including healthy, mild, moderate, severe, and progressed, according to Table 1. The suggested model beat all previous models, reaching the highest sensitivity and specificity values, yielding a final test accuracy of 97.5%. This accuracy score is lower than the results of the multi-classification because of the unbalanced data displayed in the dataset that was supplied. On several multi-class datasets, Table 5 shows the proposed model’s accuracy and precision. Considering all metrics from Figure 8, Figure 9 and Figure 10, it can be determined that the suggested model outperforms cutting-edge models and works well in binary and multi-class data classification.

## 7. Conclusions

Millions of individuals throughout the globe experience DR. Early illness identification facilitates DR prevention. In this study, we proposed the CAD-DR detection and classification method, which is completely automated. To attain the highest accuracy, the model is developed utilizing a mix of unique pre-processing steps and deep learning models. The method reduces noise, highlights lesions, and eventually enhances DR classification performance by utilizing CLAHE and MSR image preparation approaches. The innovative image processing technique suggested in this study is the CLAHE and MSR approaches, which were utilized to emphasize the areas of DR-affected photos that are particularly crucial in determining their presence. The modified deep learning model of the RDS-DR is also proposed in this research work. The model is assessed using four distinct publicly accessible datasets as well as the authors’ dataset (DR-INSIGHTS), which incorporates several datasets gathered from various sources. The most recent comparison demonstrates the suggested methodology’s higher performance in contrast to earlier models presented in the literature. It is clear from a comparison of their advantages and disadvantages that the suggested technique performs better than the currently popular methods. The suggested technique must be tested on a sizable and complicated dataset, ideally one that contains a significant number of potential DR cases, to show its effectiveness. In addition to other augmentation approaches, NASNet, MobileNet, or EfficientNet analysis of fresh datasets may be performed in the future.

## Figures and Tables

**Figure 1 diagnostics-13-03116-f001:**
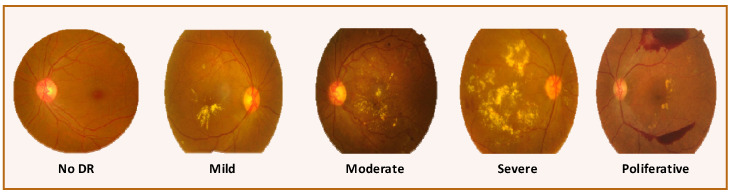
Categorization of DR images according to severity level.

**Figure 2 diagnostics-13-03116-f002:**
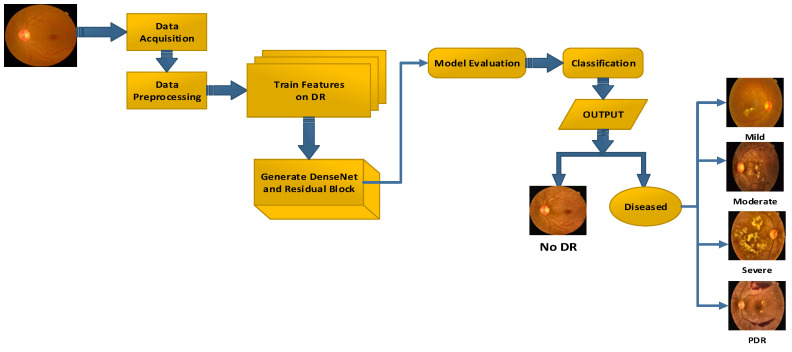
Workflow of Proposed Methodology.

**Figure 3 diagnostics-13-03116-f003:**
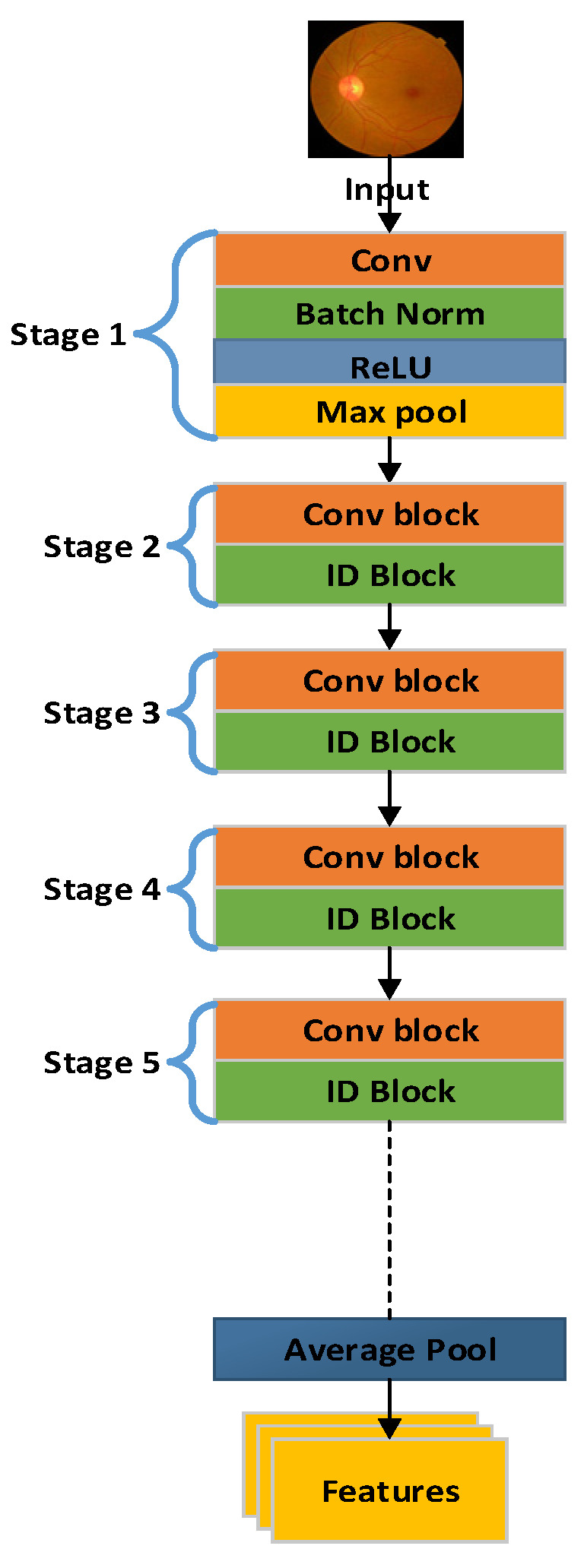
Architecture of ResNet-50.

**Figure 4 diagnostics-13-03116-f004:**
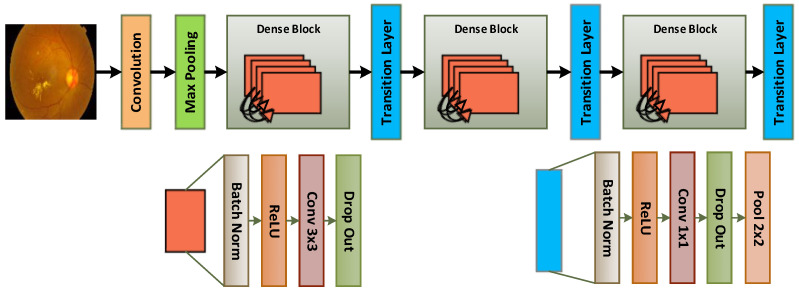
Architecture of DenseNet-121.

**Figure 5 diagnostics-13-03116-f005:**
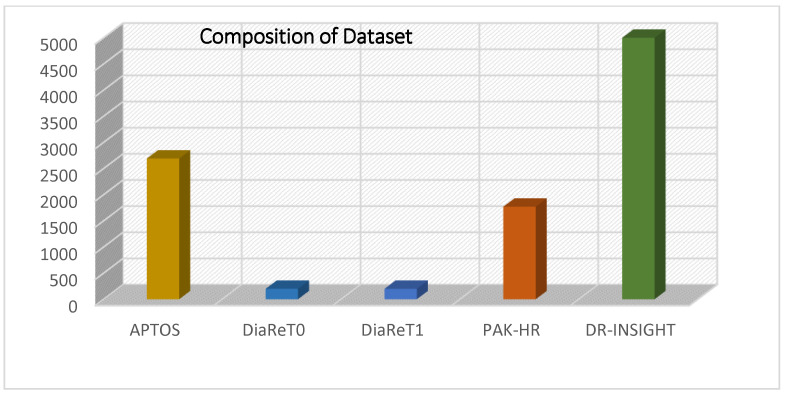
Composition of Datasets.

**Figure 6 diagnostics-13-03116-f006:**
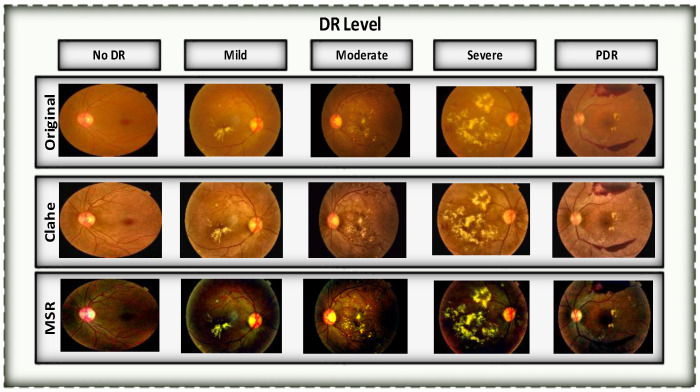
A visual result of image preprocessing to enhance contrast while adjusting noise pixels.

**Figure 7 diagnostics-13-03116-f007:**
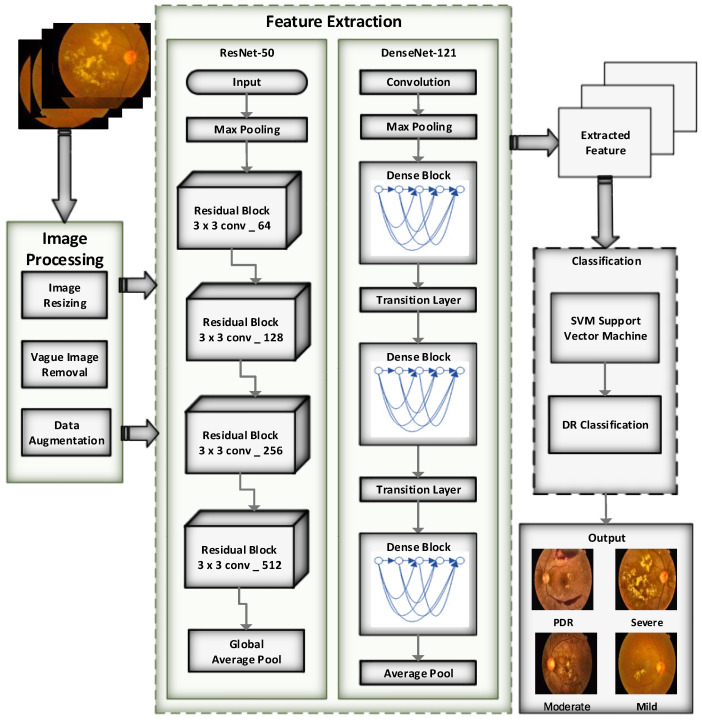
Architecture of the proposed methodology.

**Figure 8 diagnostics-13-03116-f008:**
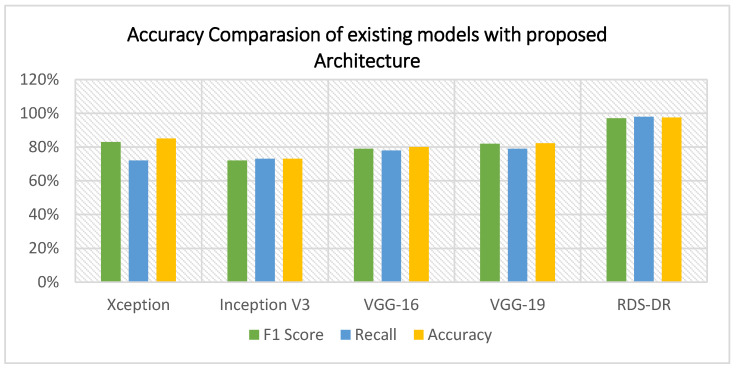
Accuracy comparison of different deep learning models.

**Figure 9 diagnostics-13-03116-f009:**
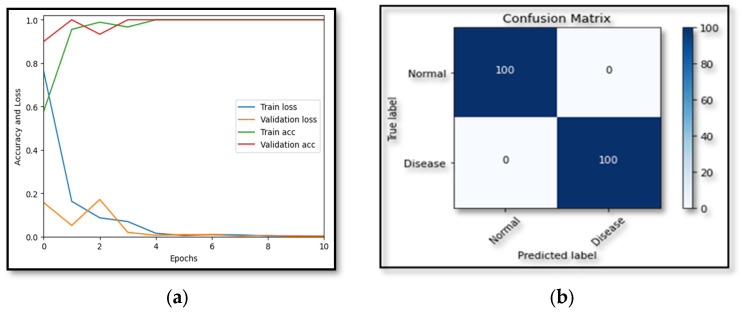
Accuracy loss along with confusion metric with HR dataset. (**a**) Shows the training and validation accuracy along with the respective loss. (**b**) Confusion matrix.

**Figure 10 diagnostics-13-03116-f010:**
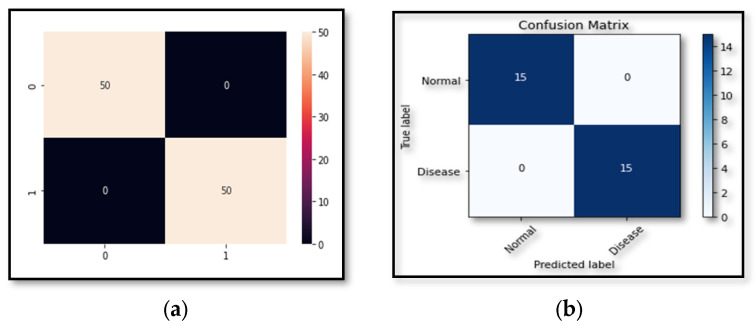
Confusion results of DiaRet0 and DiaRet1 datasets. (**a**) Shows confusion results of DiaRet0. (**b**) Shows confusion results of DiaRet1.

**Figure 11 diagnostics-13-03116-f011:**
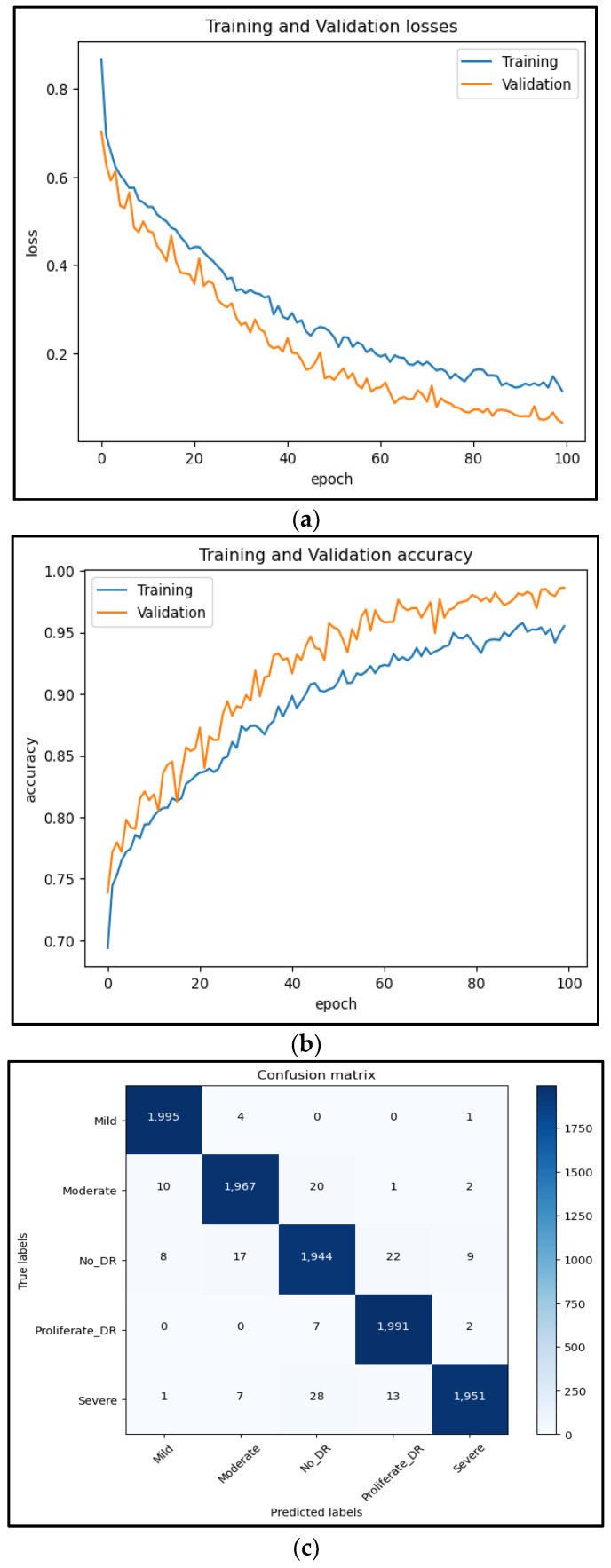
Results with APTOS dataset. (**a**) Shows the training and validation loss of the APTOS dataset. (**b**) Shows the training and validation accuracy of APTOS dataset. (**c**) Confusion matrix of the APTOS dataset.

**Figure 12 diagnostics-13-03116-f012:**
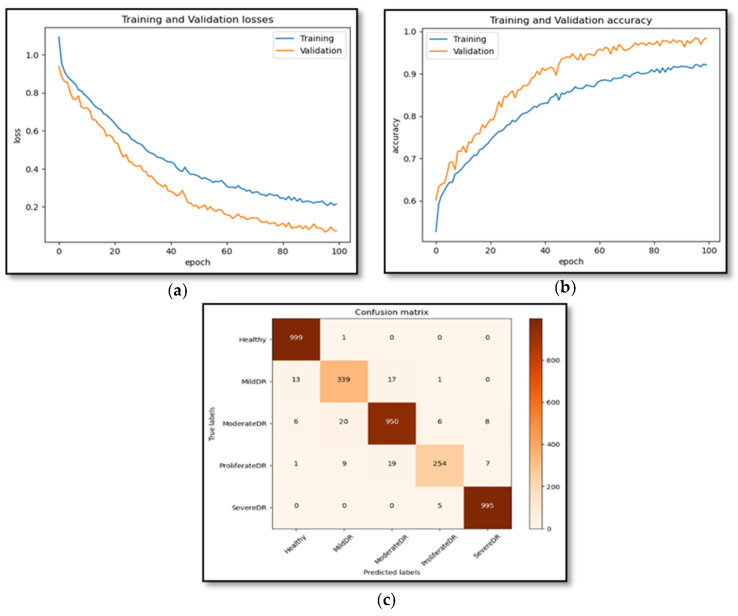
Accuracy and results of the DR-Insight dataset. (**a**) Shows the training and validation loss of the DR-Insight dataset. (**b**) Shows the training and validation accuracy of DR-Insight dataset. (**c**) Confusion matrix of DR-Insight dataset.

**Figure 13 diagnostics-13-03116-f013:**
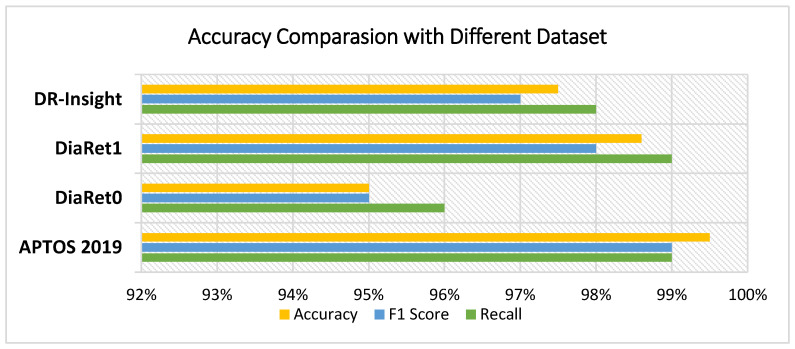
Accuracy comparison with different datasets.

**Figure 14 diagnostics-13-03116-f014:**
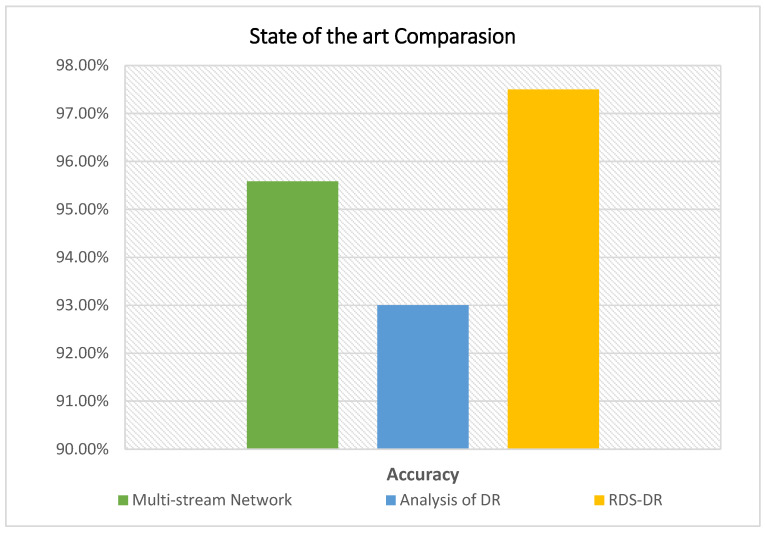
State of the art comparison with different research.

**Figure 15 diagnostics-13-03116-f015:**
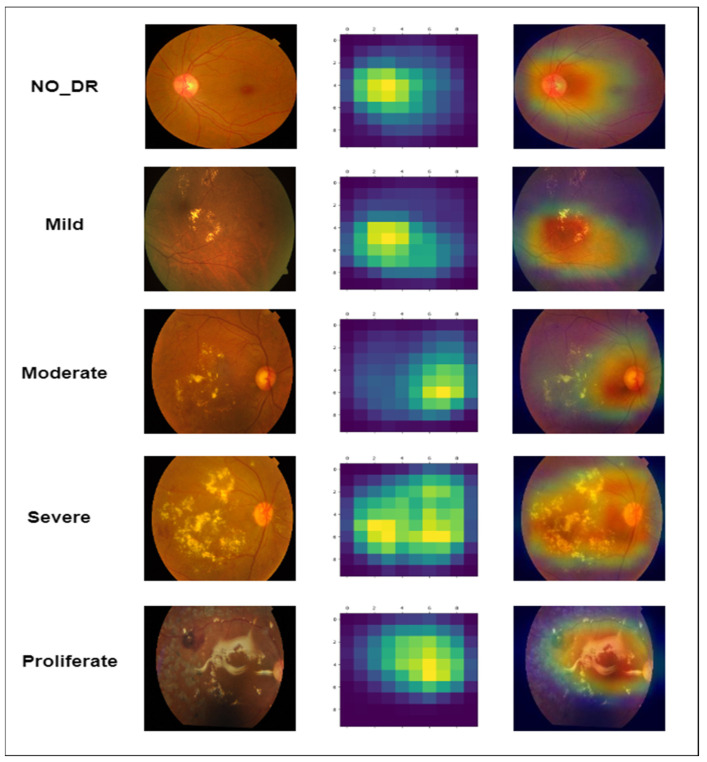
Patterns identified severity levels of the proposed RDS-DR system when diagnosed by DR images.

**Table 1 diagnostics-13-03116-t001:** DR findings as per severity level.

Retinal Image Analysis/Findins	DR Level
Microaneurysms (MAs) only	Mild
Hemorrhages	Moderate
Microvascular abnormalities	
Hard exudates	
Soft exudates	
Venous beading	
>20 intra-retinal hemorrhages	Severe NPDR
Retinal vein occlusion	
Noticeable intra-retinal MA	
Neovascularization	PDR
Vitreous/pre-retinal hemorrhage	

**Table 2 diagnostics-13-03116-t002:** Existing work for DR prediction by various former researchers.

Ref.	Year	Technique Used	Architecture	Dataset	Results
[6]	2013		GMM, SVM	DiaRet0, DiaRet1	
[7]	2014		GMM	DiaRetDB	
[8]	2013	PNN	SVM, Decision Tree		
[9]	2016	DNN	-	EyePACS-1 Messidor-2	ACC 93.4% ACC 93.9%
[10]	January 2016	CNN	-	Kaggle EyePACS	ACC 75%
[11]	2017	CNN	-	EyePACS	ACC 83.68%
[12]	2018	DCNN	AlexNet VGG16 InceptionNet V3	Kaggle dataset	ACC 37.43% ACC 50.03% ACC 63.23%
[13]	2018	DCN	ResNet	Kaggle dataset	85%
[14]	2018	BNN, DNN, CNN	VGGnet model	-	72.5%
[15]	2018	CNN	AlexNet, VggNet, GoogleNet, ResNet,	Kaggle	-
[16]	December 2018		leave-one-out method	-	ACC 80.2%
[17]	July 2019	CNN	Inception-V3, Inception-ResNet- V2, and ResNet-152	-	ACC 88.21%.
[18]	July 2019	CNN	ResNet-50, Inception-V2, Inception-V3, Xception, and DenseNet	13,767 images	-
[19]	July 2019	CNN	AlexNet	Kaggle dataset IDRiD	ACC 90.07%.
[20]	August 2019	CNN (wp-CNN).	Resnet, Se-net, and DenseNet models	-	ACC 94.23%
[21]	2019	DNN	ResNet-50, Inception-v3, Xception, DenseNet-121, Dense-169	Kaggle dataset	ACC 80%.
[22]	2020	CNN	Binary	Messidor-2 dataset	ACC 91%
[23]	2021	ADL-CNN	Path-based technique with expected gradient length based active learning method.	EyePACS dataset	ACC 98%
[24]	2020	DCNN	VGG16 and MobileNetV1		ACC 89.42% ACC 89.84%
[25]	April 2020	DNN	DenseNet121	APTOS 2019 dataset.	ACC 96.51%
[26]	April 2020	DCNN	Dense Net, ResNet50, and VGG16,		AUC 0.8140 AUC 0.7125

**Table 3 diagnostics-13-03116-t003:** Summary of recent work on DR Classification.

Author	Year	Methodology
Majumdar [27]	2021	Squeeze excitation densely connected deep CNN
Kobat [28]	2022	DenseNet201
Saranya [29]	2022	Support vector machine (SVM)
Mohamed M. Farag [30]	2022	DenseNet169 + CBAM
Hayati [31]	2023	EfficientNet
Pradeep Kumar Jena [32]	2023	U-net for segmentation CNN with SVM for classification
S. Zulaikha Beevi [33]	2023	SqueezeNet, the picture is classed into the normal or abnormal DCNN for severity level.
Al-Antary [34]	2021	Multi-scale attention network (MSA-Net)
Macsik [35]	2022	Local binary, convolutional neural network (LBCNN)

**Table 4 diagnostics-13-03116-t004:** Dataset used for training and testing purposes, where 375 patients’ data were used, averaging 15 photos per patient.

Reference	Name	Normal Image	Diseased Image	Size	No of Images
[36]	DiaReT0	100	100	(1125 × 1264) pixels	200
[36]	DiaReT1	100	100	(1125 × 1264) pixels	200
[37]	APTOS	600	2090	(1125 × 1264) pixels	2690
Private	PAK-HR	560	1210	(1125 × 1264) pixels	1770
Private	DR-Insight	1000	4000	(1125 × 1264) pixels	5000
		2360	7500	Downsizing: 700 × 600 pixels	9860

**Table 5 diagnostics-13-03116-t005:** Augmentation techniques applied on the dataset.

Augmentation Techniques	Value
Width shift range	0.2
Rotation range	15
Zoom Range	0.2
Crop	True
Vertical Flip	False
Horizontal Flip	True

**Table 6 diagnostics-13-03116-t006:** Proposed architecture comparison with a state-of-the-art model in terms of accuracy and prediction time in seconds.

Model	F1 Score	Recall	Accuracy	Prediction Time (s)
Xception	83	72	84.95%	12.23 s
Inception V3	72	73	73%	13.43 s
VGG-16	79	78	80%	11.33 s
VGG-19	82	79	82.20%	11.44 s
**RDS-DR**	**97**	**98**	**97.5%**	**5.22 s**

**Table 7 diagnostics-13-03116-t007:** Comparison of accuracy, recall, and F1 score of RDS-DR with different datasets.

Datasets	Recall	F1 Score	Accuracy
APTOS 2019	99	99	99.5%
DiaRet0	96	95	95%
DiaRet1	99	98	98.6%
**DR-Insight**	**98**	**97**	**97.5%**

**Table 8 diagnostics-13-03116-t008:** Performance comparison between Multi-Stream Network, DR, and RDS-DR.

Methods	Accuracy
Multi-stream Network	95.58%
Analysis of DR	93%
**RDS-DR**	**97.5%**

## Data Availability

We have used two publicly available datasets shared by [36,37,39], accessed on 5 July 2023. The dataset provided by [36,37] can be downloaded at https://www.kaggle.com/datasets/nguyenhung1903/diaretdb1-v21 (accessed on 5 July 2023), https://www.kaggle.com/datasets/mariaherrerot/aptos2019 and other datasets are private (accessed on 5 July 2023).

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
