# Peer review of "RDS-DR: An Improved Deep Learning Model for Classifying Severity Levels of Diabetic Retinopathy"

_diagnostics, 2023, doi:10.3390/diagnostics13193116_

Round 1

Reviewer 1 Report

The authors developed the RDS-DR models to classify the DR images, but there are several issues that are required to improved.

1.  The details about data sources, especially the private datasets in Table 3, are required to provide. Also, the ethical information and patient consents were not provide. the authors need to improve this part.

2. the contrast enhancement looks good, and how does the enhancement and preprocessing improve the model performance? the author may compare the dataset with/without enhancement in the results. Because their results are better than literatures, whether it is because of the enhancement.

3. Fig. 8 is not clear, how did the DR images serve as the input images? the residual and dense blocks process DR images simutaniously or separately or sequencely? the average pool in Line 465 should be reflected in the Fig. 8 as well.

4. How did the author realize their models? In which platform. How about the training details, like training/testing proportion, learning rate, batch size, and so on.

5.  In table 5, how about the time cost among the proposed models and convertional models?

6. In table 7 , what are the multi-stream network? what is its difference to RDS-DR? How about the analysis of DR? Overall, section 5.4 is not clear.

7. the author may consider the feature mapping to feature out where the defective regions in DR images.

Author Response

Dear reviewer, thank you for your valuable comments and suggestions. I attached the detailed response letter for you kind consideration. 

Best regards,

Dr. Imran Qureshi 

Reviewer 2 Report

The work does not take into account how accurately the previously established classification performs the evaluation. Here, certain sections were chosen. The transfer would be more successful if it was explained in more detail which version of the ETDRS or DR classification was used.

What is the addiitional information to PMID: 37627904 DOI: 10.3390/diagnostics13162645

FFA is not a routine diagnostic procedure, but is only performed in certain constellations (see AAO recommendations).
In section 4.3 it should be mentioned how the images were acquired, which camera was used and which SOPs were followed. Did the patients consent to the evaluation/study? How many patients were photographed? What was the percentage of usable images?
According to which protocol was the pre-selection of the 9860 images done on 5 levels (á 1000 images)? It is unusual that these levels were sufficiently represented. The exact numbers are not mentioned anywhere.
Why were the individual lesions not annotated? The exact process of labeling should be more precise.
The annotations at Figure 1 are very poorly placed, making it very hard to follow. Why were so many overlapping lines chosen and not simply placed the text differently?
Table 1 does not describe symptoms, but findings.
In Table 2, an additional column should indicate which and how many images are available per patient in the corresponding dataset. Image detail and quality are relevant determinants of which separators are used.

The language is not the major factor.

Author Response

(The authors gave the same response as above.)

Round 2

Reviewer 1 Report

NO further comments.